# Sobolev Training for the Neural Network Solutions of PDEs

## Abstract

Approximating the numerical solutions of partial differential equations (PDEs) using neural networks is a promising application of deep learning. The smooth architecture of a fully connected neural network is appropriate for finding the solutions of PDEs; the corresponding loss function can also be intuitively designed and guarantees the convergence for various kinds of PDEs. However, the rate of convergence has been considered as a weakness of this approach. This paper introduces a novel loss function for the training of neural networks to find the solutions of PDEs, making the training substantially efficient. Inspired by the recent studies that incorporate derivative information for the training of neural networks, we develop a loss function that guides a neural network to reduce the error in the corresponding Sobolev space. Surprisingly, a simple modification of the loss function can make the training process similar to *Sobolev Training* although solving PDEs with neural networks is not a fully supervised learning task. We provide several theoretical justifications for such an approach for the viscous Burgers equation and the kinetic Fokker–Planck equation. We also present several simulation results, which show that compared with the traditional $L^2$ loss function, the proposed loss function guides the neural network to a significantly faster convergence. Moreover, we provide the empirical evidence that shows that the proposed loss function, together with the iterative sampling techniques, performs better in solving high dimensional PDEs.

## 1 Introduction

Deep learning has achieved remarkable success in many scientific fields, including computer vision and natural language processing. In addition to engineering, deep learning has been successfully applied to the field of scientific computing. Particularly, the use of neural networks for the numerical integration of partial differential equations (PDEs) has emerged as a new important application of the deep learning.

Being a universal approximator (Cybenko, 1989; Hornik et al., 1989; Li, 1996), a neural network can approximate solutions of complex PDEs. To find the neural network solution of a PDE, a neural network is trained on a domain wherein the PDE is defined. Training a neural network comprises the following: feeding the input data through forward pass and minimizing a predefined loss function with respect to the network parameters through backward pass. In the traditional supervised learning setting, the loss function is designed to guide the neural network to generate the same output as the target data for the given input data. However, while solving PDEs using neural networks, the target values that correspond to the analytic solution are not available. One possible way to guide the neural network to produce the same output as the solution of the PDE is to penalize the neural network to satisfy the PDE itself (Sirignano & Spiliopoulos, 2018; Berg & Nyström, 2018; Raissi et al., 2019; Hwang et al., 2020).

Unlike the traditional mesh-based schemes including the finite difference method (FDM) and the finite element method (FEM), neural networks are inherently mesh-free function-approximators. Advantageously, as mesh-free function-approximators, neural networks can avoid the curse of dimensionality (Sirignano & Spiliopoulos, 2018) and approximate the solutions of PDEs on complex geometries (Berg & Nyström, 2018). Recently, Hwang et al. (2020) showed that neural networks could approximate the solutions of kinetic Fokker–Planck equations under not only various kinds

of kinetic boundary conditions but also several irregular initial conditions. Moreover, they showed that the neural networks automatically approximate the macroscopic physical quantities including the kinetic energy, the entropy, the free energy, and the asymptotic behavior of the solutions. Further issues including the inverse problem were investigated by Raissi et al. (2019); Jo et al. (2020).

Although the neural network approach can be used to solve several complex PDEs in various kinds of settings, it requires relatively high computational cost compared to the traditional mesh-based schemes in general. To resolve this issue, we propose a novel loss function using Sobolev norms in this paper. Inspired by a recent study that incorporated derivative information for the training of neural networks (Czarnecki et al., 2017), we develop a loss function that efficiently guides neural networks to find the solutions of PDEs. We prove that the $H^1$ and $H^2$ norms of the approximation errors converge to zero as our loss functions tend to zero for the 1-D Heat equation, the 1-D viscous Burgers equation, and the 1-D kinetic Fokker–Planck equation. Moreover, we show via several simulation results that the number of epochs to achieve a certain accuracy is significantly reduced as the order of derivatives in the loss function gets higher, provided that the solution is smooth. This study might pave the way for overcoming the issue of high computational cost when solving PDEs using neural networks.

The main contributions of this work are threefold: 1) We introduce novel loss functions that enable the *Sobolev Training* of neural networks for solving PDEs. 2) We prove that the proposed loss functions guarantee the convergence of neuarl networks in the corresponding Sobolev spaces although it is not a supervised learning task. 3) We empirically demonstrate the effect of *Sobolev Training* for several regression problems and the improved performances of our loss functions in solving several PDEs including the heat equation, Burgers' equation, the Fokker–Planck equation, and high-dimensional Poisson equation.

## 2 RELATED WORKS

Training neural networks to approximate the solutions of PDEs has been intensively studied over the past decades. For example, Lagaris et al. (1998; 2000) used neural networks to solve Ordinary Differential Equations (ODEs) and PDEs on a predefined set of grid points. Subsequently, Sirignano & Spiliopoulos (2018) proposed a method to solve high-dimensional PDEs by approximating the solution using a neural network. They focused on the fact that the traditional finite mesh-based scheme becomes computationally intractable when the dimension becomes high. However, because neural networks are mesh-free function-approximators, they can solve high-dimensional PDEs by incorporating mini-batch sampling. Furthermore, the authors showed the convergence of the neural network to the solution of quasilinear parabolic PDEs under certain conditions.

Recently, Raissi et al. (2019) reported that one can use observed data to solve PDEs using physics-informed neural networks (PINNs). Notably, PINNs can solve a supervised regression problem on observed data while satisfying any physical properties given by nonlinear PDEs. A significant advantage of PINNs is that the data-driven discovery of PDEs, also called the inverse problem, is possible with a small change in the code. The authors provided several numerical simulations for various types of nonlinear PDEs including the Navier–Stokes equation and Burgers' equation. The first theoretical justification for PINNs was provided by Shin et al. (2020), who showed that a sequence of neural networks converges to the solutions of linear elliptic and parabolic PDEs in $L^2$ sense as the number of observed data increases. There also exists a study aiming to enhance the convergence of PINNs (van der Meer et al., 2020).

Additionally, several works related deep neural networks with PDEs but not by the direct approximation of the solutions of PDEs. For instance, Long et al. (2018) attempted to discover the hidden physics model from data by learning differential operators. A fast, iterative PDE-solver was proposed by learning to modify each iteration of the existing solver (Hsieh et al., 2019). A deep backward stochastic differential equation (BSDE) solver was proposed and investigated in Weinan et al. (2017); Han et al. (2018) for solving high-dimensional parabolic PDEs by reformulating them using BSDE.

The main strategy of the present study is to leverage derivative information while solving PDEs via neural networks. The authors of Czarnecki et al. (2017) first proposed *Sobolev Training* that uses derivative information of the target function when training a neural network by slightly modifying

the loss function. They showed that *Sobolev Training* had lower sample complexity than regular training, and therefore it is highly efficient in many applicable fields, such as regression and policy distillation problems. We appropriate the concept of *Sobolev Training* to develop a loss function for the efficient training of a neural network for solving PDEs.

## 3 LOSS FUNCTION

We consider the following Cauchy problem of PDEs:

$$Pu = f, \ (t,x) \in [0,T] \times \Omega, \tag{3.1}$$

$$Iu = g, \ (t,x) \in \{0\} \times \Omega, \tag{3.2}$$

$$Bu = h, \ (t,x) \in [0,T] \times \partial\Omega, \tag{3.3}$$

where $P$ denotes a differential operator; $I$ and $B$ denote the initial and boundary operators, respectively; $f$, $g$, and $h$ denote the inhomogeneous term, and initial and boundary data, respectively. In most studies that reported the neural network solutions of PDEs, a neural network was trained on uniformly sampled grid points $\{(t_i, x_j)\}_{i,j=1}^{N_t, N_x} \in [0,T] \times \Omega$, which were completely determined before training. One of the most intuitive ways to make the neural network satisfy PDEs (3.1)–(3.3) is to minimize the following loss functional:

$$Loss(u_{nn}; p) = \|Pu_{nn} - f\|_{L^p([0,T]\times\Omega)}^p + \|Iu_{nn} - g\|_{L^p(\Omega)}^p + \|Bu_{nn} - h\|_{L^p([0,T]\times\partial\Omega)}^p,$$

where $u_{nn}$ denotes the neural network and $p = 1$ or $2$, as they have been the most commonly used exponents in regression problems in previous studies. Evidently, an analytic solution $u$ satisfies $Loss(u) = 0$, and thus one can conceptualize a neural network that makes $Loss(u_{nn}) = 0$ a possible solution of PDEs (3.1)–(3.3). This statement is in fact proved for second-order parabolic equations with the Dirichlet boundary condition in Jo et al. (2020), and for the Fokker–Planck equation with inflow and specular reflective boundary conditions in Hwang et al. (2020). Both the proofs are based on the following inequality:

$$\|u - u_{nn}\|_{L^\infty(0,T;L^2(\Omega))} \le C Loss(u_{nn}; 2),$$

for some constant $C$, which states that minimizing the loss functional implies minimizing the approximation error.

The main concept behind *Sobolev Training* is to minimize the error between the output and the target function, and that between the derivatives of the output and those of the target function. However, unlike the traditional supervised regression problem, neither the target function nor its derivative is provided while solving PDEs via neural networks. Thus, a special treatment is required to apply *Sobolev Training* for solving PDEs using neural networks. In this and the following sections, we propose several loss functions and prove that they guarantee the convergence of the neural network to the solution of a given PDE in the corresponding Sobolev space. Therefore, the proposed loss functions play similar roles to those in *Sobolev Training*.

We define the loss function that depends on the Sobolev norm $W^{k,p}$ as follows:

$$Loss_{GE}(u_{nn}; k, p, l, q) = \left\| \|P(u_{nn}(t, \cdot)) - f(t, \cdot)\|_{W^{l,q}(\Omega)}^q \right\|_{W^{k,p}([0,T])}^p, \tag{3.4}$$

$$Loss_{IC}(u_{nn}; l, q) = \|Iu_{nn}(t, x) - g(x)\|_{W^{l,q}(\Omega)}^q, \tag{3.5}$$

$$Loss_{BC}(u_{nn}; k, p, l, q) = \left\| \|Bu_{nn}(t, \cdot) - h(t, \cdot)\|_{W^{l,q}(\partial\Omega)}^q \right\|_{W^{k,p}([0,T])}^p. \tag{3.6}$$

**Remark 3.1.** *Here,* $Loss_{TOTAL}^{(0)}(u_{nn}) = Loss_{GE}(u_{nn}; 0, 2, 0, 2) + Loss_{IC}(u_{nn}; 0, 2) + Loss_{BC}(u_{nn}; 0, 2, 0, 2)$ *coincides with the traditional $L^2$ loss function employed by Sirignano & Spiliopoulos (2018); Berg & Nyström (2018); Raissi et al. (2019); Hwang et al. (2020).*

When we train a neural network, the loss functions (3.4)–(3.6) are computed by Monte-Carlo approximation. Because the grid points are uniformly sampled, the loss functions are approximated as follows:

$$Loss_{GE}(u_{nn}; k, p, l, q) \approx \frac{T|\Omega|}{N_t N_x} \sum_{|\beta| \le k} \sum_{i=1}^{N_t} \left| \frac{d^\beta}{dt^\beta} \sum_{|\alpha| \le l} \sum_{j=1}^{N_x} |D^\alpha P(u_{nn}(t_i, x_j)) - D^\alpha f(t_i, x_j)|^q \right|^p,$$

$$Loss_{IC}(u_{nn}; l, q) \approx \frac{|\Omega|}{N_x} \sum_{|\alpha| \leq l} \sum_{j=1}^{N_x} |D^\alpha u_{nn}(0, x_j) - D^\alpha g(x_j)|^q,$$

$$Loss_{BC}(u_{nn}; k, p, l, q) \approx \frac{T|\partial\Omega|}{N_t N_B} \sum_{|\beta| \leq k} \sum_{i=1}^{N_t} \left| \frac{d^\beta}{dt^\beta} \sum_{|\alpha| \leq l} \sum_{x_j \in \partial\Omega} |D^\alpha u_{nn}(t_i, x_j) - D^\alpha h(t_i, x_j)|^q \right|^p,$$

where $\alpha$ and $\beta$ denote the conventional multi-indexes, and $D$ denotes the spatial derivatives.

## 4 THEORETICAL RESULTS

In this section, we theoretically validate our claim that our loss functions guarantee the convergence of the neural network to the solution of a given PDE in the corresponding Sobolev spaces, and that they play a similar role to those in *Sobolev Training* while solving PDEs via neural networks. Throughout this section, we will denote the strong solution of each equation by $u$, neural network solution by $u_{nn}$, and Sobolev spaces $W^{1,2}$ and $W^{2,2}$ by $H^1$ and $H^2$, respectively. All the proofs are provided in the Appendix.

### 4.1 THE HEAT EQUATION AND BURGERS' EQUATION

We define the following three total loss functions for the heat equation and Burgers' equation:

$$Loss_{TOTAL}^{(0)}(u_{nn}) = Loss_{GE}(u_{nn}; 0, 2, 0, 2) + Loss_{IC}(u_{nn}; 0, 2) + Loss_{BC}(u_{nn}; 0, 2, 0, 2), \tag{4.1}$$

$$Loss_{TOTAL}^{(1)}(u_{nn}) = Loss_{GE}(u_{nn}; 0, 2, 0, 2) + Loss_{IC}(u_{nn}; 1, 2) + Loss_{BC}(u_{nn}; 0, 2, 0, 2), \tag{4.2}$$

$$Loss_{TOTAL}^{(2)}(u_{nn}) = Loss_{GE}(u_{nn}; 1, 2, 0, 2) + Loss_{IC}(u_{nn}; 2, 2) + Loss_{BC}(u_{nn}; 0, 2, 0, 2). \tag{4.3}$$

We then obtain the following convergence theorem:

**Theorem 4.1.** *(Proofs are provided in (A.5) for the heat equation, and (A.8) for Burgers' equation) For the following 1-D heat and Burgers' equations:*

| The heat equation | Burgers' equation |
|---|---|
| $u_t - u_{xx} = 0$ in $(0, T] \times \Omega$, | $u_t + uu_x - \nu u_{xx} = 0$ in $(0, T] \times \Omega$, |
| $u(0, x) = u_0(x)$ on $\Omega$, | $u(0, x) = u_0(x)$ on $\Omega$, |
| $u(t, x) = 0$ on $[0, T] \times \partial\Omega$, | $u(t, x) = 0$ on $[0, T] \times \partial\Omega$, |

*there hold, provided that $u_{nn}$ is smooth,*

$$\max_{0 \leq t \leq T} \|u(t) - u_{nn}(t)\|_{L^2(\Omega)} \to 0 \text{ as } Loss_{TOTAL}^{(0)} \to 0,$$

$$\operatorname*{ess\,sup}_{0 \leq t \leq T} \|u(t) - u_{nn}(t)\|_{H_0^1(\Omega)} \to 0 \text{ as } Loss_{TOTAL}^{(1)} \to 0,$$

$$\operatorname*{ess\,sup}_{0 \leq t \leq T} \|u(t) - u_{nn}(t)\|_{H^2(\Omega)} \to 0 \text{ as } Loss_{TOTAL}^{(2)} \to 0.$$

### 4.2 THE FOKKER–PLANCK EQUATION

For the Fokker–Planck equation, we need additional parameters for a new input variable $v$. We define the following two total loss functions for the Fokker–Planck equation:

$$Loss_{TOTAL}^{(0;FP)}(u_{nn}) = Loss_{GE}(u_{nn}; 0, 2, 0, 2, 0, 2) + Loss_{IC}(u_{nn}; 0, 2, 0, 2)$$
$$+ Loss_{BC}(u_{nn}; 0, 2, 0, 2, 0, 2), \tag{4.4}$$

$$Loss_{TOTAL}^{(1;FP)}(u_{nn}) = Loss_{GE}(u_{nn}; 0, 2, 1, 2, 1, 2) + Loss_{IC}(u_{nn}; 1, 2, 1, 2)$$
$$+ Loss_{BC}(u_{nn}; 0, 2, 0, 2, 0, 2). \quad (4.5)$$

We then have the following convergence theorem:

**Theorem 4.2.** *(Proofs are provided in (A.10) and (A.12)) For the 1-D Fokker–Planck equation with the periodic boundary condition:*

$$u_t + vu_x - \beta(vu)_v - qu_{vv} = 0, \ for \ (t, x, v) \in [0, T] \times [0, 1] \times \mathbb{R},$$
$$u(0, x, v) = u_0(x, v), \ for \ (x, v) \in [0, 1] \times \mathbb{R},$$
$$\partial_{t,x,v}^{\alpha} u(t, 1, v) - \partial_{t,x,v}^{\alpha} u(t, 0, v) = 0, \ for \ (t, v) \in [0, T] \times \mathbb{R},$$

*there hold, under assumptions (A.50) and (A.51),*

$$\sup_{0 \leq t \leq T} \|u(t) - u_{nn}(t)\|_{L^2(\Omega \times [-V,V])} \to 0 \ as \ Loss_{TOTAL}^{(0;FP)} \to 0,$$

$$\sup_{0 \leq t \leq T} \|u(t) - u_{nn}(t)\|_{H^1(\Omega; L^2([-V,V]))} \to 0 \ as \ Loss_{TOTAL}^{(1;FP)} \to 0.$$

**Remark 4.3.** *The theorems in this section imply that the proposed loss functions guarantee the convergence of neural networks in the corresponding Sobolev spaces, thereby coinciding with the main idea of Sobolev Training.*

**Remark 4.4.** *The theorems in this section cannot be directly generalized to the high-dimensional cases because even the 2-dimensional case starts involving the convexity of the boundary. Though it has also been shown that the Fokker-Planck operator has strong hypoellipticity and the solutions to the boundary problems are smooth even in the higher dimensional case, the proof requires long rigorous mathematical analysis. For more information, see Hwang et al. (2018; 2019).*

**Remark 4.5.** *Because we cannot access the label (which corresponds to the analytic solution) on the interior grid, solving PDEs using a neural network is not a fully supervised problem. Interestingly, by incorporating derivative information in the loss function, the proposed approach enables Sobolev Training even if neither the labels nor the derivatives of the target function are provided.*

## 5 EXPERIMENTAL RESULTS

In this section, we provide experimental results for toy examples that comprise several regression problems and various kinds of differential equations, including the heat equation, Burgers' equation, the kinetic Fokker–Planck equation, and high-dimensional Poisson's equation. We employ a fully connected neural network, which is a natural choice for function approximation. We use the hyperbolic tangent function as a nonlinear activation function. Although $ReLU(x) = \max(0, x)$ is a frequent choice in modern machine learning, it is not appropriate for solving PDEs because the second derivatives of the neural network vanish.

In appreciation of Automatic Differentiation, we can easily compute derivatives of any order of a neural network with respect to input data despite the compositional structure; see Baydin et al. (2017) and references therein. We implemented our neural network using PyTorch, a widely used deep learning library (Paszke et al., 2019). For the numerical experiments, we used a neural network with three hidden layers each of which had $d$-256-256-256-1 neurons, where $d$ denotes the input dimension. We used the ADAM optimizer (Kingma & Ba, 2014), a popular gradient-based optimizer.

To see whether our loss functions performed more efficiently than the traditional $L^2$ loss function introduced in Remark 3.1, we made everything maintain the same except the loss function. We compared the loss functions on the basis of $L^2(\Omega)$ test error for the toy examples, absolute relative test error for the high-dimensional Poisson equation, and $L^\infty(0, T; L^2(\Omega))$ test error for the other PDEs. For each loss function, we recorded the number of epochs required to meet a certain error threshold and the test error. Considering the randomness due to network initialization, we repeated the training a hundred times. Conversely, we initialized a hundred different neural networks with uniform initialization and trained them in the same manner. To compute the test error, we used analytic solutions for the Heat equation, Burgers' equation, and the high-dimensional Poisson equation, and a numerical solution from Wollman & Ozizmir (2008) for the kinetic Fokker–Planck equation.

## 5.1 Toy examples

First, we consider two simple regression problems with target functions $\sin(x)$ and $ReLU(x)$, respectively. For these toy examples, we define the loss functions as follows:

$$\text{L2 loss} = \|u_{nn}(x) - y(x)\|_2^2,$$
$$\text{H1 loss} = \|u_{nn}(x) - y(x)\|_2^2 + \|u'_{nn}(x) - y'(x)\|_2^2,$$
$$\text{H2 loss} = \|u_{nn}(x) - y(x)\|_2^2 + \|u'_{nn}(x) - y'(x)\|_2^2 + \|u''_{nn}(x) - y''(x)\|_2^2,$$

where $y(x)$ denotes either $\sin(x)$, or $ReLU(x)$. We uniformly sampled a hundred grid points from $[0, 2\pi]$ for training $\sin(x)$. Similarly, we uniformly sampled a hundred grid points from $[-1, 1]$ for training $ReLU(x)$. We expected the training to become fast using higher order derivatives as many as possible when training $\sin(x)$ and $ReLU(x)$. Figure 1 confirms our assumption to be true. Interestingly, although $ReLU(x)$ is not twice weakly differentiable at only one point $x = 0$, the H2 loss does not facilitate the training.

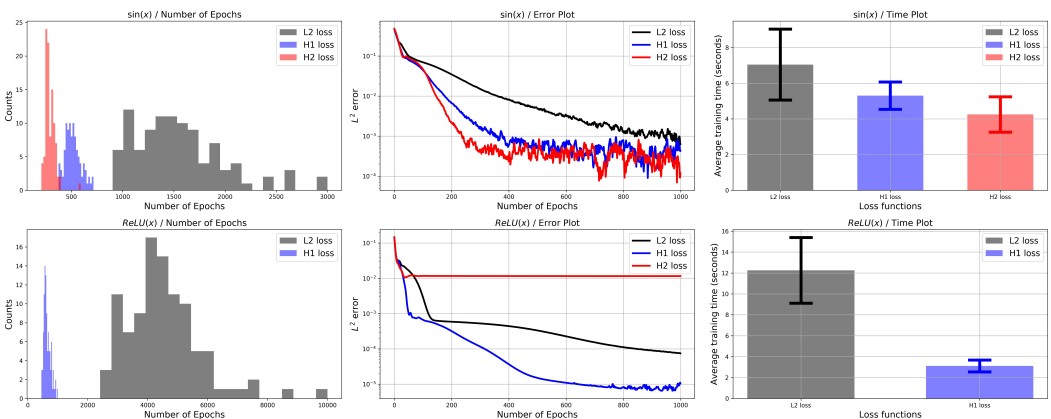

Figure 1: First row: results for $\sin(x)$, Second row: results for $ReLU(x)$. First column: Histograms generated from the repeated training of neural networks for training $\sin(x)$, and $ReLU(x)$. Second column: Test $L^2$ errors. Third column: Average training time for each loss function to achieve certain error threshold. Error bars are for standard deviations. The thresholds for the error are set to $10^{-4}$.

In order to explore the nature of *Sobolev Training*, we design more complicated toy examples. Consider the target functions $\sin(kx)$, for $k = 1, 2, ..., 5$, and $ReLU(kx) = \max(0, kx)$, for $k = 1, 2, 3..., 10$. As $k$ increases, the target functions and their derivatives contain drastic changes in their values, so it is difficult to learn those functions. We hypothesize that in *Sobolev Training*, the training becomes faster since we give explicit label for the derivatives and it becomes easier to capture the drastic changes in the derivatives. This is empirically shown to be true in Figure 2. We train neural networks to approximate $\sin(kx)$, and $ReLU(kx)$ for different $k$ and record the number of training epochs to achieve certain error threshold which can be regarded as a difficulty of the problem. As one can see in Figure 2, the difficulty changes little to no when we train with H1 and H2 losses while the difficulty increases with $k$ when L2 loss is used. This implies that the difficulty of training barely changes in *Sobolev Training* even the target function has stiff changes. The same observations are made when solving PDEs. The improvement of our loss functions compare to L2 loss function are more dramatic for Burgers' equation (which has stiff solution (Raissi et al., 2019)) than for the heat equation, with the initial condition of $f_2$ (which has a higher frequency) than with the initial condition of $f_1$ initial condition in the Fokker–Planck equation, and as $k$ increases for the high-dimensional Poisson equation 7.

## 5.2 The heat equation & Burgers' equation

We now demonstrate the results of the *Sobolev Training* of the neural networks for solving PDEs. We begin with the 1-D heat equation, and Burgers' equation, which is the simplest PDE that combines both the nonlinear propagation effect and diffusive effect. Burgers' equation often appears as a

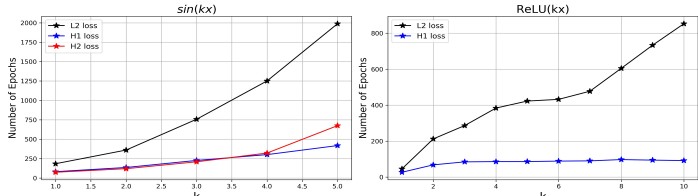

Figure 2: Average number of epochs to make error less than $10^{-3}$ increases in L2 loss as k increases. However, when we use H1, and H2 losses, required number of epochs increases much more slowly or stays the same as k increases.

simplification of a more complex and sophisticated model, such as the Navier–Stokes equation. The equations with the homogeneous Dirichlet boundary condition read as follows:

| The Heat equation | Burgers' equation |
|---|---|
| $u_t - u_{xx} = 0$ in $(0,10] \times [0,\pi]$, | $u_t + uu_x - 0.2u_{xx} = 0$ in $(0, 0.01] \times [0,1]$, |
| $u(0,x) = \sin(x)$ on $[0,\pi]$, | $u(0,x) = -\sin(\pi x)$ on $[0,1]$, |
| $u(t,x) = 0$ on $[0,10] \times \{0,\pi\}$. | $u(t,x) = 0$ on $[0,0.01] \times \{0,1\}$. |

The heat equation attains a unique analytic solution $u(t,x) = \sin(x)\exp(-t)$; an analytic solution of Burgers' equation is provided in Basdevant et al. (1986).

Although Sirignano & Spiliopoulos (2018) indicated that iterative random sampling reduces the computational cost, we fixed the grid points before training because we aimed to compare the efficiency of our loss function with that of the traditional one. For the heat equation and the Burgers' equation, we uniformly sampled the grid points $\{t_i, x_j\}_{i,j=1}^{N_t, N_x}$ from $(0,T] \times \Omega$, where $N_t$ and $N_x$ denote the number of samples for interior $t$ and $x$, respectively. For the initial and boundary conditions, we sampled the grid points from $\{t = 0, x_j\}_{j=1}^{N_x} \in \{0\} \times \Omega$ and $\{t_i, x_j\}_{i,j=1}^{N_t, N_B} \in [0,T] \times \partial\Omega$, respectively, where $N_B$ denotes the number of grid points in $\partial\Omega$. Here, we set $N_t, N_x, N_B = 31$. The testing data were also uniformly sampled from the domain of the PDEs.

The L2, H1, and H2 losses are the Monte-Carlo approximations of (4.1), (4.2), and (4.3), respectively, for the heat equation and Burgers' equation. Working on achieving a smooth solution, we observed that the H2 loss performed the best, followed by the H1 loss and then the L2 loss in both accuracy, and computation time. We show the corresponding results in Figure 3.

## 5.3 THE FOKKER–PLANCK EQUATION

The kinetic Fokker–Planck equation describes the dynamics of a particle whose behavior is similar to that of the Brownian particle. The Fokker–Planck operator has a strong regularizing effect not just in the velocity variable but also in the temporal and the spatial variables by the hypoellipticity. The Fokker–Planck equation has been considered in numerous physical circumstances including the Brownian motion described by the Uhlenbeck-Ornstein processes.

We provide two simulation results for different initial conditions for the 1-D Fokker–Planck equation with the periodic boundary condition. For the Fokker–Planck equation, we adopted the idea of sampling from Hwang et al. (2020). Because it is practically difficult to consider the entire space for the $v \in \mathbb{R}$ variable, we truncated the space for $v$ as $[-5, 5]$. We then uniformly sampled the grid points $\{t_i, x_j, v_k\}_{i,j,k=1}^{N_t, N_x, N_v}$ from $(0,T] \times \Omega \times [-5,5]$, where $N_v$ denotes the number of samples for $v$. The grid points for the initial and periodic boundary conditions were accordingly sampled. The truncated equation reads as follows:

$$u_t + vu_x - \beta(vu)_v - qu_{vv} = 0, \text{ for } (t,x,v) \in (0,3] \times [0,1] \times [-5,5],$$
$$u(0,x,v) = f(x,v), \text{ for } (x,v) \in [0,1] \times [-5,5],$$
$$\partial_{t,x,v}^\alpha u(t,1,v) - \partial_{t,x,v}^\alpha u(t,0,v) = 0, \text{ for } (t,v) \in [0,3] \times [-5,5],$$

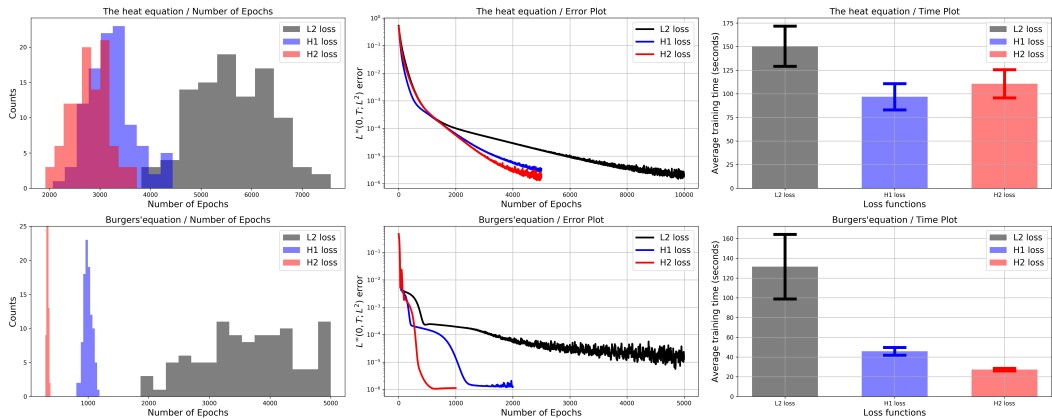

Figure 3: First row: results for the heat equation. Second row: results for Burgers' equation. First column: Histograms for the heat and Burgers' equation generated from a hundred neural networks for each loss function. Second column: Test $L^\infty(0, T; L^2(\Omega))$ errors. Third column: Average training time for each loss function to achieve certain error threshold. Error bars are for standard deviations. The thresholds for the error are set to $10^{-5}$.

where $f(x, v)$ is either $f_1(x, v) = \frac{\exp(-v^2)}{\int_{-5}^{5} \exp(-v^2) dv}$, or $f_2(x, v) = \frac{(1+\cos(2\pi x)) \exp(-v^2)}{\int_0^1 \int_{-5}^5 (1+\cos(2\pi x)) \exp(-v^2) dv dx}$, and $\beta = 0.1, q = 0.1$.

A numerical solution on the test data was computed by a method shown by Wollman & Ozizmir (2008) and used for computing the test error. L2 loss and H1 loss denote the Monte-Carlo approximations of (4.4) and (4.5), respectively. The values of $N_t, N_x$, and $N_v$ were set to be 31, and the grid points were uniformly sampled. Expectedly, a solution of the Fokker–Planck equation could be estimated substantially faster using our loss function in both cases. We have provided the detailed results in Figure 4.

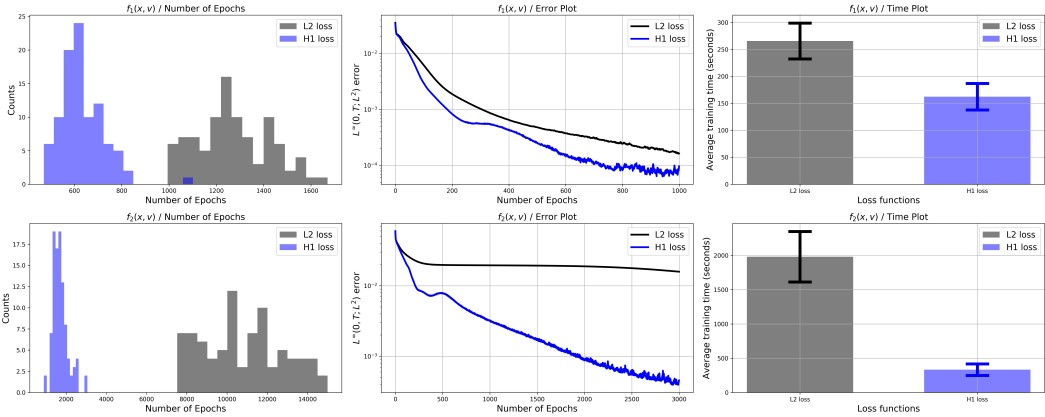

Figure 4: First row: results for $f_1$ initial condition. Second row: results for $f_2$ initial condition. First column: Histograms generated from a hundred neural networks for each loss function. Second column: Test $L^\infty(0, T; L^2(\Omega))$ errors. Third column: Average training time for each loss function to achieve certain error threshold. Error bars are for standard deviations. The thresholds for the errors for the initial conditions $f_1(x, v)$, and $f_2(x, v)$ are set to $10^{-4}$, and $10^{-3}$, respectively.

## 5.4 THE HIGH-DIMENSIONAL POISSON EQUATION

The Poisson equation serves as an example problem in the recent literature; see Weinan & Yu (2018); Hsieh et al. (2019); Zang et al. (2020). In this section, we provide empirical results to demonstrate

that the proposed loss functions perform satisfactorily when equipped with iterative sampling for solving high-dimensional PDEs; see Sirignano & Spiliopoulos (2018) for more information. Convergence result similar to those of in section 4 for the Poisson equation is given in section A.4. We consider the following high-dimensional Poisson equation with the Dirichlet boundary condition:

$$- \triangle u = \frac{\pi^2}{4} \sum_{i=1}^{d} \sin(\frac{\pi}{2}x_i), \text{ for } x \in \Omega = (0,1)^d,$$

$$u = \sum_{i=1}^{d} \sin(\frac{\pi}{2}x_i), \text{ for } x \in \partial\Omega,$$

where $x = (x_1, x_2, ..., x_d) \in \Omega$. One can readily prove that $u(x) = \sum_{i=1}^{d} \sin(\frac{\pi}{2}x_i)$ is a strong solution. We compare the following three loss functions with each other:

$$Loss_{TOTAL}^{(0;Poisson)}(u_{nn}) = Loss_{GE}(u_{nn}; 0, 2) + Loss_{BC}(u_{nn}; 0, 2), \quad (5.1)$$

$$Loss_{TOTAL}^{(1;Poisson)}(u_{nn}) = Loss_{GE}(u_{nn}; 1, 2) + Loss_{BC}(u_{nn}; 0, 2), \quad (5.2)$$

$$Loss_{TOTAL}^{(2;Poisson)}(u_{nn}) = Loss_{GE}(u_{nn}; 1, 2) + Loss_{BC}(u_{nn}; 1, 2). \quad (5.3)$$

Notably, the aforementioned loss functions have the variable $x$ only. Table 1 presents the relative errors on a predefined test set for $d = 10, 50$, and 100. Evidently, in all cases, the proposed loss functions outperform the traditional $L^2$ loss function. More detailed experimental results are given in section B.

Table 1: Average of the relative errors of a hundred neural networks for the high-dimensional Poisson's equations. We uniformly sampled 500 data points from $\Omega$ for each epoch and trained the neural networks in 10000 epochs at a learning rate $10^{-4}$.

| Dimension | $Loss_{TOTAL}^{(0;Poisson)}$ | $Loss_{TOTAL}^{(1;Poisson)}$ | $Loss_{TOTAL}^{(2;Poisson)}$ |
|---|---|---|---|
| 10 | 0.38% | **0.22**% | **0.22**% |
| 50 | 2.00% | 1.74% | **1.52**% |
| 100 | 3.15% | 3.06% | **2.89**% |

## 6 DISCUSSION AND CONCLUSION

Inspired by *Sobolev Training*, we proposed novel loss functions, which efficiently guided the training of neural networks for solving PDEs. We theoretically justified that the proposed loss functions guaranteed the convergence of a neural network to a solution of PDEs in the corresponding Sobolev spaces. We also discussed that the proposed theorems imply that the training becomes *Sobolev Training* by slightly modifying the loss function, although the process of estimating neural network solutions of PDEs is not fully supervised.

In addition to the toy examples, which showed the exceptional speed of *Sobolev Training*, we provided empirical evidences demonstrate that our loss functions expedited the training more than the traditional $L^2$ loss function. We believe that this can solve the problem associated with the high costs involved in estimating the neural network solutions of PDEs. Moreover, our experiments on high-dimensional problems showed that the proposed loss function performed better when equipped with iterative grid sampling. The histograms in Figure 1-4 indicate that our loss function provided more stable training in that it reduced the variance in the distribution of the number of epochs (e.g., for the Burgers' equation, L2 loss: 3651±812, H1 loss: 995±71, and H2 loss: 331±15). Thus, the training, when governed by our loss function, became robust to the random initialization of the weights.

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

## A   Proofs for the Theorems in Section 4

We begin with the basic definitions of Sobolev spaces. We excerpt the definitions from Evans (2010).

**Definition A.1.** *Suppose $u, v \in L^1_{loc}(U)$ and $\alpha$ is a multiindex. We say that $v$ is the $\alpha^{th}$-weak derivative of $u$, written*

$$D^\alpha u = v,$$

*provided*

$$\int_U u D^\alpha \phi dx = (-1)^{|\alpha|} \int_U v\phi dx,$$

*for all test functions $\phi \in C_c^\infty(U)$.*

Now we define the Sobolev space. Fix $1 \le p \le \infty$ and let $k$ be a nonnegative integer.

**Definition A.2.** *The Sobolev space $W^{k,p}(U)$ consists of all locally integrable functions $u : U \to \mathbb{R}$ such that for each multiindex $\alpha$ with $|\alpha| \le k$, $D^\alpha$ exists in the weak sense and belongs to $L^p(U)$. Note that if $p = 2$, we usually write $H^k(U) = W^{k,2}(U)$ $(k = 0, 1, 2, ...)$.*

The Sobolev norms are defined as follow:

**Definition A.3.** *If $u \in W^{k,p}(U)$, we define its norm to be*

$$\|u\|_{W^{k,p}(U)} = \begin{cases} (\sum_{|\alpha| \le k} \int_U |D^\alpha u|^p dx)^{1/p} & (1 \le p \le \infty), \\ \sum_{|\alpha| \le k} \operatorname{ess\,sup}_U |D^\alpha u| & (p = \infty). \end{cases}$$

Finally, we define a notion of convergence in Sobolev spaces.

**Definition A.4.** *Let $\{u_m\}_{m=1}^\infty, u \in W^{k,p}(U)$. We say $u_m$ converges to $u$ in $W^{k,p}(U)$ provided*

$$\lim_{m \to \infty} \|u_m - u\|_{W^{k,p}(U)} = 0.$$

## A.1 THE HEAT EQUATION

We denote the strong solution of the heat equation

$$
\begin{aligned}
u_t - u_{xx} &= 0 \text{ in } (0,T] \times \Omega, \\
u(0,x) &= u_0(x) \text{ on } \Omega, \\
u(t,x) &= 0 \text{ on } [0,T] \times \partial\Omega,
\end{aligned}
$$

by $u$ and the neural network solution by $u_{nn}$. Then, $v = u - u_{nn}$ satisfies:

$$
\begin{aligned}
v_t - v_{xx} &= f(t,x) \text{ in } (0,T] \times \Omega, \\
v(0,x) &= g(x) \text{ on } \Omega, \\
v(t,x) &= 0 \text{ on } [0,T] \times \partial\Omega,
\end{aligned}
\tag{A.1}
$$

for some $f$, and $g$. Here, we can set the boundary to be zero by multiplying $B(x)$, where $B(x)$ is a smooth function satisfying $B(x) = \begin{cases} 0, & x \in \partial\Omega \\ \neq 0, & x \in \Omega \end{cases}$. Then the following holds:

**Theorem A.5.** *(Theorem 7.1.5 in Evans (2010)) If $g \in H^2(\Omega)$, $f_t \in L^2(0,T;L^2(\Omega))$, then,*

$$
\max_{0 \leq t \leq T} \|v(t)\|_{L^2(\Omega)} \leq C_1(\|f\|_{L^2(0,T;L^2(\Omega))} + \|g\|_{L^2(\Omega)}),
\tag{A.2}
$$

$$
\operatorname*{ess\,sup}_{0 \leq t \leq T} \|v(t)\|_{H_0^1(\Omega)} \leq C_2(\|f\|_{L^2(0,T;L^2(\Omega))} + \|g\|_{H_0^1(\Omega)}),
\tag{A.3}
$$

$$
\operatorname*{ess\,sup}_{0 \leq t \leq T} \|v(t)\|_{H^2(\Omega)} \leq C_3(\|f\|_{H^1(0,T;L^2(\Omega))} + \|g\|_{H^2(\Omega)}),
\tag{A.4}
$$

*for some $C_1, C_2, C_3$.*

By applying above theorem to (A.1), we get the results of Theorem 4.1.

**Remark A.6.** *The left hand sides in (A.2) - (A.4) are the errors of neural networks in corresponding norms, and the right hand sides are the losses (4.1) - (4.3) for the heat equation, respectively. This implies that the proposed loss functions are the upper bounds of the errors in the Sobolev spaces, and by minimizing them, we can expect the effect of Sobolev Training when solving PDEs with neural networks.*

In the rest of this section, we will show the similar results for Burgers' equation and the Fokker–Planck equation.

## A.2 BURGERS' EQUATION

We consider the strong solution $u$ of the following Burgers equation in a bounded interval $\Omega = [a,b]$,

$$
\begin{aligned}
\partial_t u + u \partial_x u - \partial_x^2 u &= 0 \quad \text{in } \Omega, & \text{(A.5)} \\
u &= 0 \quad \text{on } \partial\Omega, & \text{(A.6)}
\end{aligned}
$$

and the corresponding neural network solution $u_{nn}$ satisfying

$$
\begin{aligned}
\partial_t u_{nn} + u_{nn} \partial_x u_{nn} - \partial_x^2 u_{nn} &= f \quad \text{in } \Omega, & \text{(A.7)} \\
u_{nn} &= 0 \quad \text{on } \partial\Omega. & \text{(A.8)}
\end{aligned}
$$

with the inital data $u(0,\cdot)$ and $u_{nn}(0,\cdot)$, respectively.

The following proposition ensures the existence of a strong solution to the initial boundary value problem (A.5)–(A.6) (see Benia & Sadallah (2016)). Here, we multiply $B(x)$ to $u_{nn}(t,x)$ in order to meet the boundary condition. We use the notation $\lesssim$ where the relation $A \lesssim B$ stands for $A \leq CB$, where $C$ denotes a generic constant.

**Proposition A.7.** *(Theorem 1.2 in Benia & Sadallah (2016)) Let $u_0 \in H_0^1$. Then there exists a time $T^* = T^*(u_0) > 0$ such that the problem (A.5)–(A.6) with initial data $u_0$ has a unique solution of $u$ satisfying*

$$
\begin{aligned}
u &\in L^2(0,T^*;H^2(\Omega)) \cap C([0,T^*);H_0^1(\Omega)), \\
u_t &\in L^2(0,T^*;L^2(\Omega)).
\end{aligned}
$$

*Furthermore, if $T^* < \infty$, then $\|u\|_{H^1(\Omega)} \to \infty$ as $t \to T^*$.*

We will show that the following theorem holds.

**Theorem A.8.** *Let $u$ and $u_{nn}$ be strong solutions of (A.5)–(A.6) and (A.7)–(A.8) respectively, on the time interval $[0, T]$. For $w := u_{nn} - u$, following statements are valid.*

**(1)** There exists a continuous function

$$F_0 = F_0 \left( \|w(0, \cdot)\|_2^2, \int_0^T \|f\|_2^2 dt, \int_0^T \|\partial_x u\|_2^2 dt \right)$$

such that

$$\sup_{0 \le t \le T} \|w\|_2^2 + \int_0^T \|\partial_x w\|_2^2 dt \le F_0 \to 0,$$

as

$$\|w(0, \cdot)\|_2^2, \int_0^T \|f\|_2^2 dt \to 0.$$

**(2)** There exists a continuous function

$$F_1 = F_1 \left( \|w(0, \cdot)\|_{H^1}^2, \int_0^T \|f\|_2^2 dt, \int_0^T \|\partial_x u\|_2^2 dt \right)$$

such that

$$\sup_{0 \le t \le T} \|w\|_{H^1}^2 + \int_0^T \|\partial_x w\|_{H^1}^2 + \|\partial_t w\|_2^2 dt \le F_1 \to 0, \tag{A.9}$$

as

$$\|w(0, \cdot)\|_{H^1}^2, \int_0^T \|f\|_2^2 dt \to 0.$$

**(3)** There exists a continuous function

$$F_2 = F_2 \left( \|w(0, \cdot)\|_{H^2}^2, \int_0^T \|f\|_2^2 + \|\partial_t f\|_2^2 dt, \sup_{0 \le t \le T} \|\partial_x u\|_2^2 + \int_0^T \|\partial_t u\|_2^2 dt \right)$$

such that

$$\sup_{0 \le t \le T} \left( \|w\|_{H^2}^2 + \|w_t\|_2^2 \right) + \int_0^T \|\partial_x w\|_{H^2}^2 dt \le F_2 \to 0, \tag{A.10}$$

as

$$\|w(0, \cdot)\|_{H^2}^2, \int_0^T \|f\|_2^2 + \|\partial_t f\|_2^2 dt \to 0.$$

**Remark A.9.** *By the Morrey's embedding theorem and the Poincare's inequality, for $f \in H_0^1(\Omega)$, we have the following inequality,*

$$\|f\|_\infty^2 \lesssim \|f\|_2^2 + \|f_x\|_2^2 \lesssim \|f_x\|_2^2, \tag{A.11}$$

*Throughout the proof, we widely use (A.11).*

*Proof.* Subtracting (A.5) from (A.7), we get equations of $w$ as follows.

$$w_t - w_{xx} + w w_x + w u_x + u w_x = f \quad \text{in } \Omega, \tag{A.12}$$
$$w = 0 \quad \text{on } \partial\Omega, \tag{A.13}$$
$$w(0, \cdot) = g \quad \text{in } \Omega. \tag{A.14}$$

By multiplying $w$ to (A.12) and integrating by parts in $\Omega$, we have

$$\frac{1}{2} \frac{d}{dt} \|w\|_2^2 + \|w_x\|_2^2 \tag{A.15}$$

$$= \int_\Omega fw - \int_\Omega w^2 w_x - \int_\Omega w^2 u_x - \int_\Omega uww_x + \int_{\partial\Omega} ww_x,$$

$$= \sum_{k=1}^{5} I_k^0.$$

Now we estimate the terms on the right hand side of (A.15). Applying the Young's inequality, the Hölder's inequality, the Sobolev inequality, and thd Poincare inequality, we have

$$I_1^0 \leq \|f\|_2 \|w\|_2 \lesssim \|f\|_2^2 + \epsilon \|w\|_2^2 \lesssim \|f\|_2^2 + \epsilon \|w_x\|_2^2, \tag{A.16}$$

$$I_2^0 = \int_{\partial\Omega} \frac{1}{3} w^3 = \frac{1}{3} w^3(b) - \frac{1}{3} w^3(a) = 0, \tag{A.17}$$

$$I_3^0 \leq \|w\|_\infty \|w\|_2 \|u_x\|_2 \lesssim \|w_x\|_2 \|w\|_2 \|u_x\|_2, \tag{A.18}$$

$$\lesssim \|u_x\|_2^2 \|w\|_2^2 + \epsilon \|w_x\|_2^2,$$

$$I_4^0 \leq \|u\|_\infty \|w\|_2 \|w_x\|_2 \lesssim \|u_x\|_2^2 \|w\|_2^2 + \epsilon \|w_x\|_2^2, \tag{A.19}$$

$$I_5^0 = w(b)w_x(b) - w(a)w_x(a) = 0, \tag{A.20}$$

for any small $\epsilon > 0$. Applying estimates (A.16)–(A.20) to (A.15), we have the following inequality

$$\frac{d}{dt} \|w\|_2^2 + \|w_x\|_2^2 \lesssim \|u_x\|_2^2 \|w\|_2^2 + \|f\|_2^2, \tag{A.21}$$

(A.21) and the Grönwall inequality imply that

$$\sup_{0 \leq t \leq T} \|w\|_2^2 \lesssim e^{\int_0^T \|u_x\|_2^2 dt} \left( \|g\|_2^2 + \int_0^T \|f\|_2^2 dt \right). \tag{A.22}$$

Let us denote

$$\int_0^T \|u_x\|_2^2 dt = \beta$$

Now we integrate (A.21) between 0 and $T$ and drop the term $\|w(T)\|_2^2$ on the left-hand side to obtain

$$\int_0^T \|w_x\|^2 dt \lesssim \|g\|_2^2 + \int_0^T \|u_x\|_2^2 \|w\|_2^2 + \|f\|_2^2 dt \tag{A.23}$$

$$\lesssim (\beta e^\beta + 1) \left( \|g\|_2^2 + \int_0^T \|f\|_2^2 dt \right).$$

This completes the proof of (1) of Theorem A.8.

Next, by multiplying $-w_{xx}$ to (A.12) and integrating by parts in $\Omega$, we obtain

$$\frac{1}{2} \frac{d}{dt} \|w_x\|_2^2 + \|w_{xx}\|_2^2 \tag{A.24}$$

$$= -\int_\Omega fw_{xx} + \int_\Omega ww_x w_{xx} + \int_\Omega u_x ww_{xx} + \int_\Omega uw_x w_{xx} + \int_{\partial\Omega} w_t w_x,$$

$$= \sum_{k=1}^{5} I_k^1.$$

Similarly to (A.16)–(A.20), we estimate the terms on the right hand side of (A.24).

$$I_1^1 \leq \|f\|_2 \|w_{xx}\|_2^2 \lesssim \|f\|_2^2 + \epsilon \|w_{xx}\|_2^2, \tag{A.25}$$

$$I_2^1 \leq \|w\|_\infty \|w_x\|_2 \|w_{xx}\|_2 \lesssim \|w_x\|^4 + \epsilon \|w_{xx}\|_2^2, \tag{A.26}$$

$$I_3^1 \leq \|w\|_\infty \|u_x\|_2 \|w_{xx}\|_2 \lesssim \|u_x\|_2^2 \|w_x\|_2^2 + \epsilon \|w_{xx}\|_2^2, \tag{A.27}$$

$$I_4^1 \leq \|u\|_\infty \|w_x\|_2 \|w_{xx}\|_2 \lesssim \|u_x\|_2^2 \|w_x\|_2^2 + \epsilon \|w_{xx}\|_2^2, \tag{A.28}$$

$$I_5^1 = w_t(b)w_x(b) - w_t(a)w_x(a) = 0. \tag{A.29}$$

for any small $\epsilon > 0$. Applying estimates (A.25)–(A.29) to (A.24), we have the following inequality

$$\frac{d}{dt}\|w_x\|_2^2 + \|w_{xx}\|_2^2 \lesssim (\|w_x\|_2^2 + \|u_x\|_2^2)\|w_x\|_2^2 + \|f\|_2^2. \tag{A.30}$$

It follows from (A.23), (A.30), and the Grönwall inequality that

$$\sup_{0 \le t \le T} \|w_x\|_2^2 \lesssim e^{\int_0^T \|w_x\|_2^2 + \|u_x\|_2^2 dt} \left( \|g_x\|_2^2 + \int_0^T \|f\|_2^2 dt \right) \tag{A.31}$$

$$\lesssim e^{\beta} e^{F_0} \left( \|g_x\|_2^2 + \int_0^T \|f\|_2^2 dt \right).$$

In a similar way to (A.23), there exists a function $\tilde{F}_1 = F_1(\|f\|_{L^2(0,T;L^2)}, \|g\|_{H^1}, \beta)$ such that

$$\int_0^T \|w_{xx}\|_2^2 dt \lesssim F_1. \tag{A.32}$$

(2) of Theorem A.8 follows from (A.31), (A.32), and the fact that

$$w_t = f + w_{xx} - ww_x - wu_x - uw_x. \tag{A.33}$$

Finally, we differentiate (A.12) with respect to $t$, then we obtain

$$w_{tt} - w_{xxt} + w_t w_x + w w_{xt} + w_t u_x + w u_{xt} + u_t w_x + u w_{xt} = f_t \quad \text{in } \Omega, \tag{A.34}$$
$$w_t = 0 \quad \text{on } \partial\Omega, \tag{A.35}$$
$$w_t(0) = f + g_{xx} - gg_x - gu_{0x} - u_0 g_x \quad \text{in } \Omega. \tag{A.36}$$

By multiplying $w_t$ to (A.34) and integrating by parts in $\Omega$, we have

$$\frac{1}{2}\frac{d}{dt}\|w_t\|_2^2 + \|w_{xt}\|_2^2 \tag{A.37}$$

$$= \int_\Omega f_t w_t - \int_\Omega w_t^2 w_x - \int_\Omega w w_t w_{xt} - \int_\Omega w_t^2 u_x$$

$$- \int_\Omega w w_t u_{xt} - \int_\Omega u_t w_t w_x - \int_\Omega u w_t w_{xt} + \int_{\partial\Omega} w_t w_{xt},$$

$$= \sum_{k=1}^8 I_k^2.$$

Terms on the right hand side of (A.37) are estimated by

$$I_1^2 \le \|f_t\|_2 \|w_t\|_2 \lesssim \|f_t\|_2^2 + \epsilon\|w_{xt}\|_2^2, \tag{A.38}$$
$$I_2^2 \le \|w_t\|_\infty \|w_t\|_2 \|w_x\|_2 \lesssim \|w_x\|_2^2 \|w_t\|_2^2 + \epsilon\|w_{xt}\|_2^2, \tag{A.39}$$
$$I_3^2 \le \|w\|_\infty \|w_t\|_2 \|w_{xt}\|_2 \lesssim \|w_x\|_2^2 \|w_t\|_2^2 + \epsilon\|w_{xt}\|_2^2, \tag{A.40}$$
$$I_4^2 \le \|w_t\|_\infty \|w_t\|_2 \|u_x\|_2 \lesssim \|u_x\|_2^2 \|w_t\|_2^2 + \epsilon\|w_{xt}\|_2^2, \tag{A.41}$$
$$I_5^2 + I_6^2 = \int_\Omega w w_{xt} u_t \le \|w\|_\infty \|u_t\|_2 \|w_{xt}\|_2 \lesssim \|u_t\|_2^2 \|w_t\|_2^2 + \epsilon\|w_{xt}\|_2^2, \tag{A.42}$$
$$I_7^2 \le \|u\|_2 \|w_t\|_2 \|w_{xt}\|_2 \lesssim \|u_x\|_2^2 \|w_t\|_2^2 + \epsilon\|w_{xt}\|_2^2, \tag{A.43}$$
$$I_8^2 = 0. \tag{A.44}$$

for any small $\epsilon > 0$. Applying estimates (A.38)–(A.44) to (A.37), we have the following inequality

$$\frac{d}{dt}\|w_t\|_2^2 + \|w_{xt}\|_2^2 \lesssim \left( \|w_x\|_2^2 + \|u_x\|_2^2 + \|u_t\|_2^2 \right) \|w_t\|_2^2 + \|f_t\|_2^2. \tag{A.45}$$

It follows from (A.36), (A.45) and the Grönwall inequality that

$$\sup_{0 \le t \le T} \|w_t\|_2^2 \lesssim e^{\int_0^T \|w_x\|_2^2 + \|u_x\|_2^2 + \|u_t\|_2^2 dt} \left( \|w_t(0)\|_2^2 + \int_0^T \|f_t\|_2^2 \right) \tag{A.46}$$

$$\lesssim e^{\gamma} e^{F_0} \left( \|f_0\|_2^2 + \|g_{xx}\|_2^2 + \|g_x\|_2^4 + \|u_{0x}\|_2^2 \|g_x\|_2^2 + \int_0^T \|f_t\|_2^2 dt \right).$$

where

$$\gamma = \int_0^T \|u_x\|_2^2 + \|u_t\|_2^2 dt.$$

In a similar way to the proof of (2) of Theorem A.8, (3) of Theorem A.8 follows from (A.33) and (A.46). This completes the proof of the Theorem. $\qquad\square$

### A.3 THE FOKKER–PLANCK EQUATION

#### A.3.1 BOUNDARY LOSS DESIGN

Define the loss function for the periodic boundary condition as

$$
\begin{aligned}
Loss_{BC} \\
&= \sum_{|\alpha|=1} \int_0^T dt \int_{-5}^5 dv \left| \partial_{t,x,v}^\alpha f^{nn}(t,1,v;m,w,b) - \partial_{t,x,v}^\alpha f^{nn}(t,0,v;m,w,b) \right|^2 \\
&\approx \frac{1}{N_{i,k}} \sum_{|\alpha|=1, i, k} \left| \partial_{t,x,v}^\alpha f^{nn}(t_i,1,v_k;m,w,b) - \partial_{t,x,v}^\alpha f^{nn}(t_i,0,v_k;m,w,b) \right|^2 . \quad \text{(A.47)}
\end{aligned}
$$

#### A.3.2 THE FOKKER–PLANCK EQUATION IN A PERIODIC INTERVAL

In this section, we introduce an $L^2$ energy method for the Fokker–Planck equation and introduce a regularity inequality for the solutions to the equation. Throughout the section, we will abuse the notation and use both notations $\partial_z u$ and $u_z$ for the same derivative of $u$ with respect to $z$.

We consider the Fokker–Planck equation in a periodic interval $[0,1]$:

$$
\begin{aligned}
u_t + v u_x - \beta(vu)_v - q u_{vv} = 0, \text{ for } (t,x,v) \in [0,T] \times [0,1] \times \mathbb{R}, \\
u(0,x,v) = u_0(x,v), \text{ for } (x,v) \in [0,1] \times \mathbb{R}, \text{ and} \\
\partial_{t,x,v}^\alpha u(t,1,v) - \partial_{t,x,v}^\alpha u(t,0,v) = 0, \text{ for } (t,v) \in [0,T] \times \mathbb{R},
\end{aligned} \quad \text{(A.48)}
$$

for any 3-dimensional multi-index $\alpha$ such that $|\alpha| \leq 1$ and a given initial distribution $u_0 = u_0(x,v)$. Now we consider the Fokker–Planck equation that the corresponding neural network solution $u_{nn}$ would satisfy:

$$
\begin{aligned}
(u_{nn})_t + v(u_{nn})_x - \beta(vu_{nn})_v - q(u_{nn})_{vv} = f \text{ for } (t,x,v) \in [0,T] \times [0,1] \times [-5,5], \\
u_{nn}(0,x,v) = g, \text{ for } (x,v) \in [0,1] \times [-5,5], \\
\sum_{|\alpha|=1} \int_0^T dt \int_{-5}^5 dv \left| (\partial_{t,x,v}^\alpha u_{nn})(t,1,v) - (\partial_{t,x,v}^\alpha u_{nn})(t,0,v) \right|^2 \leq L,
\end{aligned} \quad \text{(A.49)}
$$

for any 3-dimensional multi-index $\alpha$ such that $|\alpha| \leq 1$ and given $f = f(t,x,v)$, $g = g(x,v)$, and a constant $L > 0$. Suppose that $f$, $g$ and $h$ are $C^1$ functions. Also, we suppose that the a priori solutions $u$ and $u_{nn}$ are sufficiently smooth; indeed, we require them to be in $C_{t,x,v}^{1,1,2}$.

For the a priori solution $u$ and $u_{nn}$ to equation A.48 and equation A.49, assume that if $|v|$ is sufficiently large, then we have that for some sufficiently small $\epsilon > 0$,

$$\sup_{t \in [0,T]} \left\| \partial_{t,x,v}^\alpha u(t,\cdot,\pm 5) - \partial_{t,x,v}^\alpha u_{nn}(t,\cdot,\pm 5) \right\|_{L_x^2([0,1])} \leq \epsilon, \quad \text{(A.50)}$$

for $|\alpha| \leq 1$ and $\alpha = (0,0,2)$. Also, suppose that

$$\left| \partial_{t,x,v}^\alpha u(t,x,\pm 5) \right|, \left| \partial_{t,x,v}^\alpha u_{nn}(t,x,\pm 5) \right| \leq C, \quad \text{(A.51)}$$

for some $C < \infty$ for $|\alpha| \leq 1$ and $\alpha = (0,0,2)$. Now we introduce the following theorem on the energy estimates:

**Theorem A.10.** *Let $u$ and $u_{nn}$ be the classical solutions to equation A.48 and equation A.49, respectively. Then we have*

$$\sup_{0 \leq t \leq T} \|u_{nn}(t) - u(t)\|_2^2 + 2(q - \varepsilon) \int_0^T \|\partial_v u_{nn}(s) - \partial_v u(s)\|_2^2 ds$$

$$\leq \left( \|g - u_0\|_2^2 + \frac{L}{2} \right) \exp\left[ \left( 1 + \frac{25\beta^2}{2\varepsilon} \right) T \right] + \int_0^T \|f(s)\|_2^2 ds + 2q\epsilon CT,$$

*for any $\varepsilon \in (0, q)$, where $L, u_0, f, g, \beta, q, m, \epsilon$, and $C$ are given in equation A.48-equation A.51.*

*Proof.* Define $w \stackrel{\text{def}}{=} u_{nn} - u$. Then by equation A.48 and equation A.49, $w$ satisfies

$$w_t + vw_x - \beta vw_v - qw_{vv} = f \text{ for } (t, x, v) \in [0, T] \times [0, 1] \times [-5, 5],$$
$$w(0, x, v) = w_0, \text{ for } (x, v) \in [0, 1] \times [-5, 5], \quad \text{(A.52)}$$

where $w_0 \stackrel{\text{def}}{=} g - u_0$. By multiplying $w$ to equation A.52 and integrating with respect to $dxdv$, we have

$$\frac{1}{2}\frac{d}{dt} \iint_{[0,1] \times [-5,5]} |w|^2 dxdv + \iint_{[0,1] \times [-5,5]} vw_x w dxdv - \iint_{[0,1] \times [-5,5]} qw_{vv} w dxdv$$

$$= \iint_{[0,1] \times [-5,5]} fw dxdv + \iint_{[0,1] \times [-5,5]} \beta vw_v w dxdv.$$

Then we take the integration by parts and obtain that

$$\frac{1}{2}\frac{d}{dt} \iint_{[0,1] \times [-5,5]} |w|^2 dxdv + \frac{1}{2} \int_{-5}^5 dv\, v(w(t, 1, v)^2 - w(t, 0, v)^2)$$

$$+ q \iint_{[0,1] \times [-5,5]} |w_v|^2 dxdv$$

$$= \iint_{[0,1] \times [-5,5]} fw dxdv + \iint_{[0,1] \times [-5,5]} \beta vw_v w dxdv$$

$$+ q \int_{[0,1]} w_v(t, x, 5)w(t, x, 5)dx - q \int_{[0,1]} w_v(t, x, -5)w(t, x, -5)dx$$

$$\stackrel{\text{def}}{=} I_1 + I_2 + I_3 + I_4.$$

We first define

$$A(t) \stackrel{\text{def}}{=} \left| \frac{1}{2} \int_{-5}^5 dv\, v(w(t, 1, v)^2 - w(t, 0, v)^2) \right|.$$

We now estimate $I_1$-$I_4$ on the right-hand side. By the Hölder inequality and Young's inequality, we have

$$|I_1| \leq \|f\|_2 \|w\|_2 \leq \frac{1}{2}\|f\|_2^2 + \frac{1}{2}\|w\|_2^2,$$

where we denote

$$\|h\|_2 \stackrel{\text{def}}{=} \iint_{[0,1] \times [-5,5]} |h|^2 dxdv.$$

Similarly, we observe that

$$|I_2| \leq 5\beta \|w_v\|_2 \|w\|_2 \leq \varepsilon \|w_v\|_2^2 + \frac{25\beta^2}{4\varepsilon} \|w\|_2^2,$$

for a sufficiently small $\varepsilon > 0$ as $|v| \leq 5$. By equation A.50, we have

$$|I_3 + I_4| \leq q\|w_v(t, \cdot, 5)\|_{L_x^2} \|w(t, \cdot, 5) - w(t, \cdot, -5)\|_{L_x^2}$$
$$+ q\|w_v(t, \cdot, 5) - w_v(t, \cdot, -5)\|_{L_x^2} \|w(t, \cdot, -5)\|_{L_x^2} \leq 2q\epsilon C.$$

Altogether, we have

$$\frac{d}{dt}\|w\|_2^2 + 2(q-\varepsilon)\|w_v\|_2^2 \leq \|f\|_2 + \left(1 + \frac{25\beta^2}{2\varepsilon}\right)\|w\|_2^2 + A(t) + 2q\epsilon C.$$

We integrate with respect to the temporal variable on $[0, t]$ and obtain

$$\|w(t)\|_2^2 + 2(q-\varepsilon)\int_0^t \|w_v(s)\|_2^2 ds$$
$$\leq \|w(0)\|_2^2 + \int_0^t \left(\|f(s)\|_2 + \left(1 + \frac{25\beta^2}{2\varepsilon}\right)\|w(s)\|_2^2 + A(s) + 2q\epsilon C\right) ds.$$

By equation A.49$_3$, we have

$$\int_0^t A(s)ds \leq \frac{L}{2}.$$

Thus, by the Grönwall inequality, we have

$$\|w(t)\|_2^2 + 2(q-\varepsilon)\int_0^t \|w_v(s)\|_2^2 ds$$
$$\leq \left(\|w_0\|_2^2 + \frac{L}{2}\right)\exp\left[\left(1 + \frac{25\beta^2}{2\varepsilon}\right)t\right] + \int_0^t \|f(s)\|_2^2 ds + 2q\epsilon Ct,$$

where $w_0(x, v) = g(x, v) - u_0(x, v)$. This completes the proof for the theorem. $\square$

Regarding the derivatives $\partial_t w$ and $\partial_x w$ we can obtain the similar estimates as follows.

**Corollary A.11.** *Let $u$ and $u_{nn}$ be the classical solutions to equation A.48 and equation A.49, respectively. Assume that equation A.51 holds. Then for $z = t$ or $x$ we have*

$$\sup_{0 \leq t \leq T} \|\partial_z u_{nn}(t) - \partial_z u(t)\|_2^2 + 2(q-\varepsilon)\int_0^T \|\partial_v \partial_z u_{nn}(s) - \partial_v \partial_z u(s)\|_2^2 ds$$
$$\leq \left(\|\partial_z g - \partial_z u_0\|_2^2 + \frac{L}{2}\right)\exp\left[\left(1 + \frac{25\beta^2}{2\varepsilon}\right)T\right] + \int_0^T \|\partial_z f(s)\|_2^2 ds + 2q\epsilon CT,$$

*for any $\varepsilon \in (0, q)$, where $L, u_0, f, g, \beta, q, m, \epsilon$, and $C$ are given in equation A.48-equation A.51.*

*Proof.* For both $\partial_z = \partial_t$ and $\partial_x$, we take $\partial_z$ onto equation A.52 and obtain

$$(\partial_z w)_t + v(\partial_z w)_x - \beta v(\partial_z w)_v - q(\partial_z w)_{vv} = \partial_z f \text{ for } (t, x, v) \in [0, T] \times [0, 1] \times [-5, 5],$$
$$(\partial_z w)(0, x, v) = (\partial_z w)_0, \text{ for } (x, v) \in [0, 1] \times [-5, 5],$$
$$\text{(A.53)}$$

where $(\partial_z w)_0 \overset{\text{def}}{=} \partial_z g - \partial_z u_0$. Then the proof is the same as the one for Theorem A.10 for $\partial_z w$ replacing the role of $w$. This completes the proof. $\square$

Finally, we can also obtain the regularity estimates for the derivative $\partial_v w$ as follows:

**Theorem A.12.** *Let $u$ and $u_{nn}$ be the classical solutions to equation A.48 and equation A.49, respectively. Assume that equation A.51 holds. Then we have*

$$\sup_{0 \leq t \leq T} \|\partial_v u_{nn}(t) - \partial_v u(t)\|_2^2 + 2(q-\varepsilon)\int_0^T \|\partial_{vv} u_{nn}(s) - \partial_{vv} u(s)\|_2^2 ds$$
$$\leq (L + \|\partial_x g - \partial_x u_0\|_2^2 + \|\partial_v g - \partial_v u_0\|_2^2)\exp\left[\left(2 + \frac{25\beta^2}{2\varepsilon}\right)T\right]$$
$$+ \int_0^T (\|\partial_x f(s)\|_2^2 + \|\partial_v f(s)\|_2^2)ds + 4q\epsilon CT,$$

*for any $\varepsilon \in (0, q)$, where $L, u_0, f, g, \beta, q, m, \epsilon$, and $C$ are given in equation A.48-equation A.51.*

*Proof.* we take $\partial_v$ onto equation A.52 and obtain

$$(\partial_v w)_t + w_x + v(\partial_v w)_x - \beta v(\partial_v w)_v - q(\partial_v w)_{vv} = \partial_v f,$$
$$\text{for } (t, x, v) \in [0, T] \times [0, 1] \times [-5, 5], \tag{A.54}$$

where $(\partial_v w)_0 \stackrel{\text{def}}{=} \partial_v g - \partial_v u_0$. By multiplying $\partial_v w$ to equation A.54 and integrating with respect to $dxdv$, we have

$$\frac{1}{2}\frac{d}{dt}\iint_{[0,1]\times[-5,5]} |\partial_v w|^2 dxdv + \iint_{[0,1]\times[-5,5]} v(\partial_v w)_x (\partial_v w) dxdv$$

$$- \iint_{[0,1]\times[-5,5]} q(\partial_v w)_{vv} \partial_v w dxdv$$

$$= \iint_{[0,1]\times[-5,5]} (-w_x + \partial_v f)\partial_v w dxdv + \iint_{[0,1]\times[-5,5]} \beta v(\partial_v w)_v \partial_v w dxdv.$$

Then we take the integration by parts and obtain that

$$\frac{1}{2}\frac{d}{dt}\iint_{[0,1]\times[-5,5]} |w_v|^2 dxdv$$

$$+ \frac{1}{2}\int_{[-5,5]} \left(v(\partial_v w)^2(t,1,v) - v(\partial_v w)^2(t,0,v)\right) dv$$

$$+ q\iint_{[0,1]\times[-5,5]} |w_{vv}|^2 dxdv$$

$$= \iint_{[0,1]\times[-5,5]} \partial_v f w_v dxdv + \iint_{[0,1]\times[-5,5]} \beta v w_{vv} w_v dxdv$$

$$+ q\int_{[0,1]} w_{vv}(t,x,5)w_v(t,x,5)dx - q\int_{[0,1]} w_{vv}(t,x,-5)w_v(t,x,-5)dx$$

$$- \iint_{[0,1]\times[-5,5]} w_x w_v dxdv \stackrel{\text{def}}{=} I_1 + I_2 + I_3 + I_4 + I_5.$$

We first define

$$B(t) \stackrel{\text{def}}{=} \left| \frac{1}{2}\int_{-5}^{5} dv\, v(\partial_v w(t,1,v)^2 - \partial_v w(t,0,v)^2) \right|.$$

We now estimate $I_1$-$I_4$ on the right-hand side. We now estimate $I_1$-$I_5$ on the right-hand side. By the Hölder inequality and Young's inequality, we have

$$|I_1| \le \|\partial_v f\|_2 \|w_v\|_2 \le \frac{1}{2}\|\partial_v f\|_2^2 + \frac{1}{2}\|w_v\|_2^2,$$

where we denote

$$\|h\|_2 \stackrel{\text{def}}{=} \iint_{[0,1]\times[-5,5]} |h|^2 dxdv.$$

Similarly, we observe that

$$|I_2| \le 5\beta \|w_{vv}\|_2 \|w_v\|_2 \le \varepsilon\|w_{vv}\|_2^2 + \frac{25\beta^2}{4\varepsilon}\|w_v\|_2^2,$$

for a sufficiently small $\varepsilon > 0$ as $|v| \le 5$. By equation A.50 and equation A.51, we have

$$|I_3 + I_4| \le q\|w_{vv}(t,\cdot,5)\|_{L_x^2}\|w_v(t,\cdot,5) - w_v(t,\cdot,-5)\|_{L_x^2}$$
$$+ q\|w_{vv}(t,\cdot,5) - w_{vv}(t,\cdot,-5)\|_{L_x^2}\|w_v(t,\cdot,-5)\|_{L_x^2} \le 2q\epsilon C.$$

Finally, we have

$$|I_5| \le \|w_x\|_2\|w_v\|_2 \le \frac{1}{2}\|w_x\|_2^2 + \frac{1}{2}\|w_v\|_2^2.$$

Altogether, we have

$$\frac{d}{dt}\|w_v\|_2^2 + 2(q-\varepsilon)\|w_{vv}\|_2^2 \le \|\partial_v f\|_2 + \|w_x\|_2^2 + \left(2 + \frac{25\beta^2}{2\varepsilon}\right)\|w_v\|_2^2 + \frac{5L}{2} + 2q\epsilon C.$$

Then we take the integration with respect to the temporal variable on $[0, t]$ and obtain that

$$\|w_v(t)\|_2^2 + \int_0^t ds\, 2(q - \varepsilon)\|w_{vv}(s)\|_2^2$$

$$\leq \|w_v(0)\|_2^2 + \int_0^t ds\, \left(\|\partial_v f(s)\|_2 + \|w_x(s)\|_2^2 + \left(2 + \frac{25\beta^2}{2\varepsilon}\right)\|w_v(s)\|_2^2 + B(s) + 2q\epsilon C\right).$$

By equation A.49$_3$, we have

$$\int_0^t ds\, B(s) \leq \frac{L}{2}.$$

Thus, by the Grönwall inequality, we have

$$\|w_v(t)\|_2^2 + 2(q - \varepsilon)\int_0^t \|w_{vv}(s)\|_2^2 ds$$

$$\leq \left(\frac{L}{2} + \|\partial_v w_0\|_2^2\right) \exp\left[\left(2 + \frac{25\beta^2}{2\varepsilon}\right) t\right] + \int_0^t (\|w_x(s)\|_2^2 + \|\partial_v f(s)\|_2^2) ds + 2q\epsilon Ct, \quad \text{(A.55)}$$

where $\partial_v w_0(x, v) = g(x, v) - u_0(x, v)$. Then we use Corollary A.11 for an upper-bound of $\|\partial_x w(s)\|_2^2$ and obtain that

$$\int_0^T \|w_x(s)\|_2^2 ds$$

$$\leq \left(\frac{L}{2} + \|\partial_x g - \partial_x u_0\|_2^2\right) \exp\left[\left(1 + \frac{25\beta^2}{2\varepsilon}\right) T\right] + \int_0^T \|\partial_x f(s)\|_2^2 ds + 2q\epsilon CT.$$

Then by equation A.55, we obtain that

$$\sup_{0 \leq t \leq T} \|w_v(t)\|_2^2 + 2(q - \varepsilon)\int_0^T \|w_{vv}(s)\|_2^2 ds$$

$$\leq (L + \|\partial_x w_0\|_2^2 + \|\partial_v w_0\|_2^2) \exp\left[\left(2 + \frac{25\beta^2}{2\varepsilon}\right) T\right]$$

$$+ \int_0^T (\|\partial_x f(s)\|_2^2 + \|\partial_v f(s)\|_2^2) ds + 4q\epsilon CT,$$

where $\partial_v w_0(x, v) = \partial_v g(x, v) - \partial_v u_0(x, v)$. This completes the proof for the theorem. $\square$

## A.4 THE POISSON EQUATION

We consider the Poisson equation equation with Dirichlet boundary condition:

$$-\triangle u = f \quad \text{in } \Omega,$$
$$u = g \quad \text{on } \partial\Omega.$$

Suppose there exists

$$\tilde{g} \in H^2(\bar{\Omega}) \; s.t. \; \tilde{g}|_{\partial\Omega} = g \tag{A.56}$$

Then, the equation can be written by:

$$-\triangle v = \tilde{f} \quad \text{in } \Omega,$$
$$v = 0 \quad \text{on } \partial\Omega,$$

where $v = u - \tilde{g}, \tilde{f} = f - \triangle\tilde{g}$. Therefore, we assume the homogeneous Dirichlet boundary condition provided (A.56).

Now, let u be a strong solution of

$$-\triangle u = f \quad \text{in } \Omega,$$
$$u = 0 \quad \text{on } \partial\Omega, \tag{A.57}$$

and let $u_{nn}$ be a neural network such that

$$-\triangle u_{nn} = f_{nn} \quad \text{in } \Omega,$$
$$u_{nn} = 0 \quad \text{on } \partial\Omega. \tag{A.58}$$

Here, we can set the boundary to be zero by multiplying $B(x)$, where $B(x)$ is a smooth function satisfying $B(x) = \begin{cases} 0, & x \in \partial\Omega \\ \neq 0, & x \in \Omega \end{cases}$ . By subtracting (A.58) from (A.57), we get

$$-\triangle(u - u_{nn}) = (f - f_{nn}) \quad \text{in } \Omega,$$
$$u - u_{nn} = 0 \quad \text{on } \partial\Omega. \tag{A.59}$$

Then we apply below theorem to (A.59) to get the convergence results.

**Theorem A.13.** *(Theorem 6.3.5 in Evans (2010)) Let $m$ be a nonnegative integer. Suppose that $u \in H_0^1(\Omega)$ is a weak solution of the boundary-value problem (A.57). Assume $\partial\Omega$ is $C^{m+2}$. Then,*

$$\|u\|_{H^{m+2}(\Omega)} \leq C(\|f\|_{H^m(\Omega)} + \|u\|_{L^2(\Omega)}), \tag{A.60}$$

*Furthermore, if $u$ is the unique solution of A.57, then*

$$\|u\|_{H^{m+2}(\Omega)} \leq C\|f\|_{H^m(\Omega)}. \tag{A.61}$$

By applying (A.61) to (A.59), we obtain

$$\|u - u_{nn}\|_{H^{m+2}(\Omega)} \leq C\|f - f_{nn}\|_{H^m(\Omega)}.$$

where the right hand side corresponds to $Loss_{GE}(u_{nn}; m, 2)$.

# B ADDITIONAL EXPERIMENTAL RESULTS

## B.1 DEPENDENCY ON LEARNING RATES

In this subsection, we show several experiments that shows the proposed loss functions generally performs better in different learning rates. We first show the results for Burgers' equation. In Figure 5, we show the test errors versus training epochs plot for different learning rates. We used $10^{-3}, 10^{-4}, 10^{-5}$ as learning rates and we observe that H2 loss performs best followed by H1 and L2 loss functions.

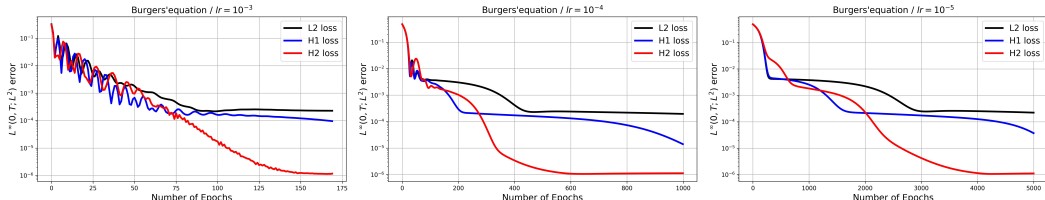

Figure 5: Test errors as training goes for different learning rates.

We next present the similar experiments for the high-dimensional Poisson equation. We trained 30 neural networks with different initializations with different learning rates. The average errors are presented in Figure 6. As the same in the Burgers' equation, our loss functions performs better than the traditional one in all learning rates.

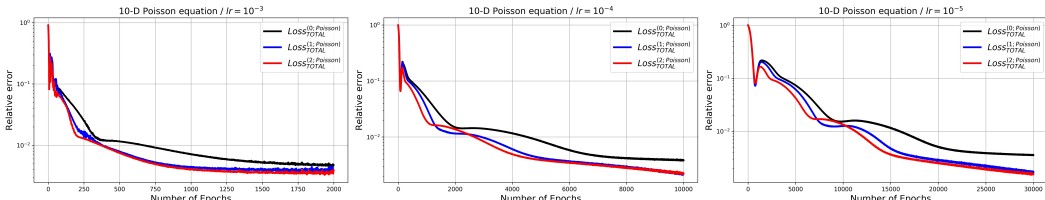

Figure 6: Test errors as training goes for different learning rates.

## B.2 HIGH-DIMENSIONAL POISSON EQUATION

In this subsection, we consider the high-dimensional Poisson equation with different boundary condition. In subsection 5.1, we pointed out that the "difficulty" of learning $\sin(kx)$ increases as $k$ increases. As a generalization of the argument, we consider the following PDEs:

$$-\triangle u = \frac{(k\pi)^2}{4} \sum_{i=1}^{d} \sin(\frac{k\pi}{2}x_i), \text{ for } x \in \Omega = (0,1)^d,$$

$$u = \sum_{i=1}^{d} \sin(\frac{k\pi}{2}x_i), \text{ for } x \in \partial\Omega,$$

for d=10, k = 1,3, and 5. As one can see in Figure 7, the improvement of Sobolev training gets bigger as $k$ increases. This observation coincides with the one in section 5, as we expected. Moreover, we present the comparison of training time to meet a certain error value for different loss functions in Figure 7. The result shows that it is advantageous to use the proposed loss functions in time, even in high-dimensional case.

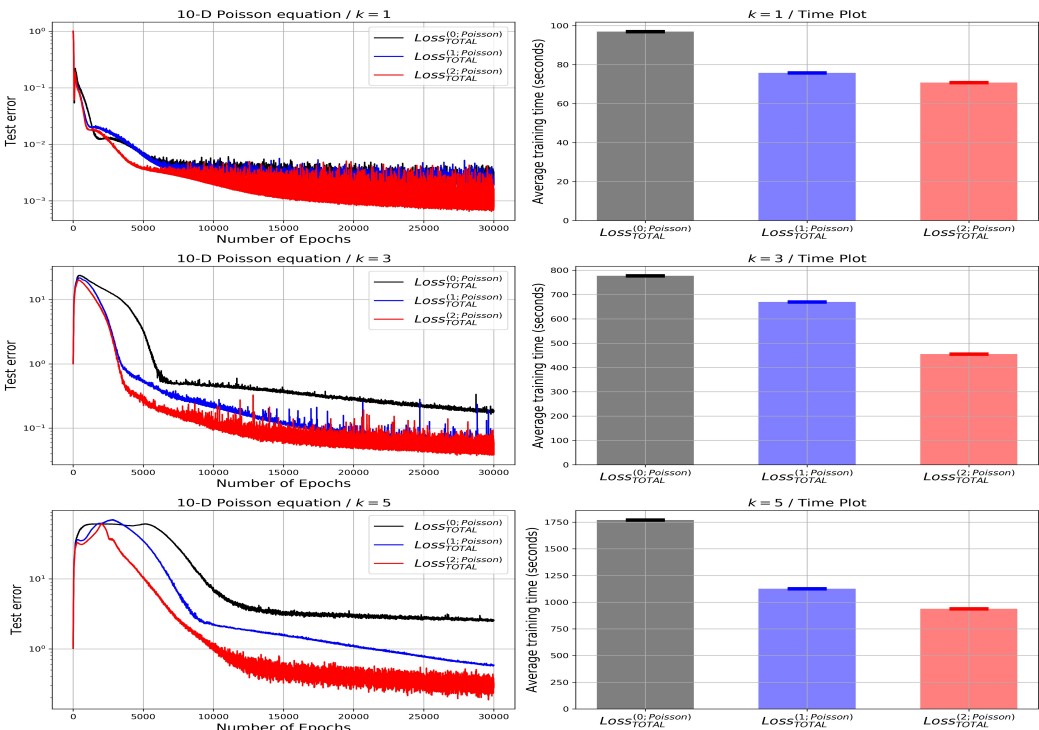

Figure 7: Left column: Test errors as training goes for different values of $k$. Right column: Required Training Time to achieve a certain test error.

