# OpenReview forum: "Sobolev Training for the Neural Network Solutions of PDEs"
_ICLR.cc/2021/Conference — Reject_

### Official Review · AnonReviewer4 · 2020-10-23
**Interesting direction but could be strengthened by more theory and numerics**

**Rating:** 4
**Confidence:** 5

**Review:**

Overview:
The paper proposes a novel loss function using Sobolev norms to decrease the computational costs when solving PDEs using neural networks. I think the idea of the work is very interesting and relevant and in a useful direction, however I think the paper in its current form is not yet suitable for publication and it should be strengthened by incorporating more theoretical and numerical aspects. This will make the concept a lot more convincing. I present some ideas and comments for improvement below.

Comments and ideas for improvement as well as clarification questions:
- You state “it requires relatively high computational cost compared to traditional mesh-based schemes in general”. Would you have a reference for this? In part I agree with the notion that neural network training could be computationally-heavy, but on the other hand, as you also mention, it should not suffer from the curse of dimensionality compared to mesh-based methods which would seem that it is computationally efficient?
- The main claim of this work is to introduce Sobolev training to speed up convergence, or as you mention in the introduction “overcoming the issue of high computational cost when solving PDEs using neural networks”. Theoretically the results in Section 4 are not showing this. I know that in the original Sobolev training paper there is a result on how Sobolev training has a lower sample complexity than regular training. Extending such a result to this setting would be necessary to make the claims in the introduction rigorous.
- The results in Thm 4.1 and 4.2 are only for 1D equations. I understand that higher order could be more complex, and perhaps the 1D equations are sufficient to convey the intuition, however in that case at least a comment is needed on how these results could be extended to higher orders.
- In Figure 1, what is the reason for H2 loss not speeding up convergence with the ReLU? Is it the differentiability?
The results in 5.2 are again only for 1D equations. I think that if theoretically you do not prove the results for high-dimensional PDEs, the value of the proposed methodology for high-dimensional PDEs should at least be shown in extensive numerical experiments. I do think the example in 5.4 is in the right direction, but a more rigorous analysis would be needed.
- I would like to have more information on how the “true” (PDE) values of the gradients of the boundary and interior differential operators are computed, and whether this is always possible.
- My last comment is a general one, but given that the research in this area is growing rapidly with various approaches to improve convergence, a stronger literature review would be necessary. I give two examples of papers which could be of interest below.

Some references which may be of interest:
- Ito, Kazufumi, Christoph Reisinger, and Yufei Zhang. "A Neural Network-Based Policy Iteration Algorithm with Global H^ 2 H 2-Superlinear Convergence for Stochastic Games on Domains." Specifically, Remark 4.2 could be of interest. The authors also discuss how certain norms cannot guarantee convergence of the derivatives of the numerical solutions.
- van der Meer, Remco, Cornelis Oosterlee, and Anastasia Borovykh. "Optimally weighted loss functions for solving PDEs with Neural Networks.”. The authors discuss the choice of loss functions to also speed up / improve convergence and the solution accuracy.

---

> ### Author Response · Authors · 2020-11-15
> **Continued**
>
>
> •	Regarding the ReLU function, we think that the differentiability matters. Consider that a neural network $u_{nn}$ learns a function $\max(0,x)$. When we minimize the $L^2$ loss function, $u_{nn}$ approximates $\max(0,x)$, and $\frac{d}{dx} u_{nn}$ has to approximate the step function $1_{x>0}$. However, since $\frac{d}{dx} u_{nn}$ is a smooth function, given the tanh activation, it is difficult to learn a step function. This procedure becomes much easier when we use the $H^1$ loss which explicitly provides a step function as a label for $\frac{d}{dx} u_{nn}$. On the other hand, if we optimize $H^2$ loss, the label for the second derivative $\frac{d^2}{dx^2} u_{nn}$ will be identically zero except for $x=0$. Therefore, $\frac{d}{dx} u_{nn}$ has to learn a step function while satisfying $\frac{d^2}{dx^2} u_{nn}$=0 and hence a conflict occurs here. We believe this is the reason for the failure of $H^2$ loss function. This inspires our new experiment in Section 5.1.
>
> •	For the high-dimensional Poisson equation, we have further included a relevant proof in section A.4 and more detailed experiments that support our claim in section B. New experiments are the generalizations of the one in section 5.1 to the PDE problem, and the results are convincing.
>
> •	As one can see in the loss functions, we don’t need the “true” (if you mean “true” an exact solution) values of the gradients of the interior differential operators. We only need to evaluate the differential operator on our neural network which can easily be done by Automatic Differentiation, and this is why we mentioned that the task is not fully-supervised. Furthermore, in this paper, we assumed that the exact form of initial & boundary conditions which is a common setting in the PDE literature. Therefore, the derivatives of initial and boundary conditions can always be computed analytically or numerically.
>
> •	We highly appreciate your suggestion for the relevant papers. We seriously reviewed the recommended works and they are reflected in our revised manuscript.
>
> [1] Justin Sirignano and Konstantinos Spiliopoulos. Dgm: A deep learning algorithm for solving partial differential equations\
> [2] Maziar Raissi, Paris Perdikaris, and George E Karniadakis.  Physics-informed neural networks:  A deep learning framework for solving forward and inverse problems involving nonlinear partial differential equations\
> [3] Lu Lu, Xuhui Meng, Zhiping Mao, George E. Karniadakis. DeepXDE: A deep learning library for solving differential equations\
> [4] Jens Berg and Kaj Nystr¨om. A unified deep artificial neural network approach to partial differential equations in complex geometries\
> [5] Wojciech M Czarnecki, Simon Osindero, Max Jaderberg, Grzegorz Swirszcz, and Razvan Pascanu. Sobolev training for neural networks\
> [6] Hyung Ju Hwang, Juhi Jang, Juan J. L. Velazquez, On the structure of the singular set for the kinetic Fokker-Planck equations in domains with boundaries. \
> [7] Hyung Ju Hwang, Juhi Jang, Jaewoo Jung, The Fokker-Planck equation with absorbing boundary conditions in bounded domains.

---

> ### Author Response · Authors · 2020-11-15
> **Response to Reviewer #4**
>
> We highly appreciate your valuable comments. We carefully read all the comments and tried to reflect the helpful comments on our revised manuscript. By doing so, there have been significant improvements in our paper. We would like to discuss the issues that you have raised as follows:
>
> •	We agree with your comment that neural networks are known to not suffering the curse of dimensionality. As pointed out in [1], neural networks can avoid the curse of dimensionality because they are mesh-free function-approximators. On the other hand, when we are dealing with low-dimensional problems, the computational cost of training a neural network is much higher compared to the mesh-based schemes. This fact discourages the use of neural networks when solving PDEs numerically in relatively low dimensions although there are many other advantages ([2,3,4]) over the mesh-based schemes. Therefore, the need for a computationally efficient algorithm for solving PDEs via neural networks naturally arises.
>
> •	What we have shown in Section 4 is the feasibility of Sobolev Training when solving PDEs via neural networks trained with the proposed loss function. Via Section 4, we first wanted to provide some theoretical support for each PDE on the convergence of the Sobolev training, before we observe the efficiency and the improvement of the Sobolev training via the numerical simulations in Seciton 5. In [5], Sobolev training is applied to the problems where both true values of the target function and its derivatives are given. However, as we have argued in Remark 4.3 and Remark 4.4, Sobolev Training can also be applied to solving PDEs via a slight modification of the loss function even if neither the target value nor its derivatives on the interior grids is provided.
> In order to further enhance the support of our claim, we conducted new experiments on the regression problems. Considering the target functions $\sin(kx)$, and $ReLU(kx) = \max(0,kx)$ with different values of $k$, we demonstrated some empirical evidence of speeding up via Sobolev Training. As $k$ increases, both the target functions and their derivatives contain drastic changes in their values so it is difficult to learn those functions (Figure 2, L2 loss). However, the number of training epochs to achieve certain error stays the same or slightly increases as k increases when we use the Sobolev loss functions while the number of epochs drastically increases in the $L2$ loss case. This implies that the difficulty of training barely changes in Sobolev training even the target function has stiff changes. We believe this can be a reason for the faster training. We summarized the results in the last paragraph of Section 5.1 and Figure 2.
>
> •	We appreciate for pointing this out. It is true that, under certain regularity assumptions on the boundary, the similar results can be derived for the higher dimensional case. In a higher dimension, it is well-known in theory that the convexity of the boundary plays a crucial role in determining the regularity of solutions for the Vlasov-Poisson and the Boltzmann kinetic equations. This is due to the fact that the flow describing the evolution of the characteristic curves is not a $C^2$ function at the singular set. On the other hand, it is well-known that solutions to the boundary value problems for the Fokker–Planck equation are smooth in all variables $t$, $x$, and $v$, even though the regularizing operator $\partial^2_v$ acts only on the velocity variable $v$. This effect is called the hypoellipticity. However, it also turns out the boundary value problems for the kinetic Fokker-Planck equation still involve nontrivial mathematical analysis on the singular sets that arise for several classes of the boundaries, and unfortunately, the analysis requires long rigorous proofs. For the mathematical literature, see [6] and [7]. Lastly, we also mention that it is also possible to consider a weak formulation in the case of the $L^2$ convergence. We have added this remark in the manuscript on Remark 4.4.

---

### Official Review · AnonReviewer3 · 2020-10-27
**Interesting problem, but incremental and not enough support for claims**

**Rating:** 5
**Confidence:** 4

**Review:**

The idea of using neural networks to approximate the solutions of the pdes is very interesting, specially in high-dimensional setting where classical approaches fail to scale. Although there has been many efforts in this direction, there are still open venues to explore. One of the most important aspect is the choice of loss function to guide the training of the neural network. And the paper's aim is to address this issue by proposing Sobolev norm as the loss function instead of the commonly used $L^2$-norm. The Sobolev norm includes additional term about derivatives of the error. The main claim is that with the inclusion of the additional term, the convergence of the neural network training becomes faster. This is the basic promise of the paper.

Although an interesting proposal, I think the paper did not address the full aspects of it. In particular,

1) training with derivatives in the loss function should be costly compared to $L^2$ loss. This becomes worse as the dimension increases. If the dimension is $d$, the first derivative scales with $d$ and the second derivative scales with $d^2$, .... It seems that at most, one can try only the first derivative.
2) No comparison is provided in terms of computational time.
3) Comparing the convergence speed may not be fair, because for $L^2$ loss, one might be able to use larger learning rate which would yield faster convergence, while large learning rate might make Sobolev training unstable.
4)  Including the derivative requires strong smoothness assumption about the pde data (the forcinig term, the BC, IC) which are not standard in the pde literature.

Given this, I think the contributions of the paper are incremental. Moreover, the theoretical results do not really support the claim that Sobolev loss function is better. It would be interesting to have a negative result about the L^2 loss function that motivates the application of the Sobolev norm.

Moreover, it seems that the appendix is not written with care.

Subsection A1 does not really include a proof. If the result is already known, it seems better to cite the reference (with exact pointer to the result) in the main body of the paper and do not include it as the contribution.

Subsection A2 also includes the result Porp A.3 without proof or reference. The proof of Thm. A4 is not written with care. The bounds in A19 and A20 are obtained without explaining the steps. What type of Poincare inequality is used?

It will be good to include a definition of the norms and Poincare inequality for the reader unfamiliar with pde analysis.

---

> ### Author Response · Authors · 2020-11-15
> **Response to Reviewer #3**
>
> We appreciate your kind and constructive comments. We have revised our manuscript in many aspects according to your comments, and there have been significant improvements in the manuscript. We would like to address below the issues that you have raised.
>
> 1.	It is true that the first derivative scales with $d$ and the second derivative scales with $d^2$. We have added the comparison of training time in Figure 7 for the high-dimensional Poisson equations. As one can see in Figure 7, the training time gets reduced when we use the proposed H1 loss even for 10-dimensional Poisson equation. Although the required training time per epoch gets larger when higher order derivatives involve, the required number of training epochs is significantly reduced. Thus, we think the H1 loss would be a good alternative of the traditional L^2 loss function even in the high dimensional problems.
> 2.	We have revised our main figures (Figure 1, 3, 4) and added Figure 7 to contain the comparison of computational times for the toy examples, heat, Burgers’, the Fokker-Planck, and the high-dimensional Poisson equation. As one can see in the revised manuscript, the computation time is significantly reduced when using the proposed loss functions in many cases of our experiments. We thank you again for the comment with the presentation of our results, which can improve the readability of our manuscript.
> 3.	We agree with you that there might be some cases that the learning rates work differently across the loss functions. Therefore we have added more experiments that are aiming to see the dependency of training on the learning rate. We train the neural networks with different loss functions and with different learning rates. We compare the results for the learning rates 1e-3, 1e-4, and 1e-5. For the Burgers equation and the high-dimensional Poisson equation, the comparison shows that the training with proposed loss functions outperforms significantly that with traditional $L^2$ loss functions, regardless of the choice of the learning rates. The new experiments are presented in the supplementary materials (section B).
> 4.	We are interested in the initial-boundary value problems for PDEs where some suitable regularity conditions are assumed on the IC and BC. In the literature of mathematical theory on the analysis of PDEs, it is also natural to assume some least amount of regularity assumptions on the initial condition; for instance, we can consider the initial condition for the Fokker-Planck equation with the notion of weak derivatives and assume that the initial condition is in $H^1_{t,x,v}$.  This assumption is sufficient for our proof in the Appendix as well. Then since the Fokker-Planck operator (or the heat operator) is hypoelliptic, we expect the regularizing effect (or the smoothing effect) on the solutions. In summary, our method can be applied even to the situation with an irregular initial condition as long as it is in $H^1_{t,x,v}$, the space of "weak" derivatives.
>
>
> Lastly, we would like to remark that we do not show the improvement on convergence or on the training time via the theoretical proofs. We emphasize that all the theoretical results in the paper have been provided as supplements, which can provide some information on the convergence of the Sobolev training. Once we check and guarantee the convergence of the Sobolev training via the theoretical arguments, we move onto the numerical experiments and check the improvement via the Sobolev training compared to the traditional L2 training. We then empirically show, in several experimental results, that the proposed loss functions work better. To better support our claim, we design a new experiment that shows why Sobolev Training makes training faster given intuition from ReLU toy example. We consider the target functions $\sin(kx)$, and $ReLU(kx)$ with different values of $k$. As $k$ increases, the target functions and their derivatives contain drastic changes in their values, so it is difficult to learn those functions (Figure 2, L2 loss). However, the number of training epochs to achieve certain errors stays the same or slightly increases as k increases when we use the Sobolev loss functions while the number of epochs drastically increases in the $L2$ loss case. This implies that the difficulty of training barely changes in Sobolev training even the target function has stiff changes. We believe this can be a reason for the faster training. We summarized the results in the last paragraph of section 5.1 and Figure 2.
>
> <Appendix section>
> We carefully revised the appendix section by adding definitions of the Sobolev norms and Sobolev spaces for the readers who are not familiar with PDE analysis as the reviewer kindly recommended. References for the proofs in A1 (A.5 in the revised version), and proposition A.3 (A.7 in the revised version) are added and we stated the inequality in Remark A.9 that we have used.

---

### Official Review · AnonReviewer1 · 2020-10-30
**applying Sobolev training to the PDE learning problem**

**Rating:** 7
**Confidence:** 4

**Review:**

Sobolev training of neural networks, which augments the standard loss function with terms that penalize discrepancies between the derivatives of the network and target functions, has been shown empirically to improve data-efficiency. Intuitively, one would expect that it also aids generalization in settings where the target function is sufficiently smooth. This manuscript proposes augmenting the loss functions used to represent the solutions of partial differential equations with terms penalizing the Sobolev norm of the solution, its initial condition, and the boundary condition. The motivation for this approach is clear because data -efficiency is of the utmost importance in PDE learning problems where the data could be very difficult to access.

The experiments in this paper clearly show that a target accuracy can be achieved with fewer overall training points when using Sobolev training, very much consistent with the established understanding of the effect of penalizing the Sobolev norms in typical supervised machine learning problems.  Some of the examples are high-dimensional and non-trivial.

The theoretical results are not particularly compelling, but they serve a reasonable justification for the proposed scheme.

---

> ### Author Response · Authors · 2020-11-15
> **Response to Reviewer #1**
>
> We thank you for taking interest in our paper and for positive comments. As all three reviewers commonly pointed out, our theoretical results show the convergence in the corresponding Sobolev spaces, but not really show the speedup of the convergence by our loss function. To overcome this, we designed in our revision new experiments that empirically show some evidence of faster training via Sobolev loss functions. We summarized the results in Section 5.1. It will be really helpful if you have any further comments.

---

### Author Response · Authors · 2020-11-15
**Common Response**

We thank all the reviewers for the constructive and valuable comments.

We responded to all the comments and we all were very interested in communicating with reviewers during the first round of the revision. The comments are really constructive and there have been vast improvement in our manuscript. Major changes of the revised manuscript are:
1.	We have added new experiments that show some evidence for speedup of convergence via Sobolev Training. (Section 5.1)
2.	We have added a theoretical justification that has similar results to the theorems in section 4, for the high-dimensional Poisson equation. (Appendix A.4)
3.	We have added several simulation results with different learning settings. We compared the loss functions for different learning rates for Burgers’ and the high-dimensional Poisson equation. We also compared the improvement of the Sobolev training via a high-dimensional Poisson equation with different boundary conditions that are natural generalizations of experiments in section 5.1 to the PDE problem. (Appendix B).

We are always open to and waiting for more discussions and comments!

---

### Decision · Program_Chairs · 2021-01-07
**Final Decision**

**Decision:**

Reject

**Comment:**

The paper solves a PDE using an additional penalty function between the derivatives of the function. On toy examples and two PDEs it is shown that these additional terms help.

Pros: - The motivation is to include derivatives in the computationa
          - Implementation and testing on several examples, including high-dimensional ones
          - Timing is included in the latest version


Cons: -The loss is Sobolev norm of the residuals of the equation.
           - The usage of the norm of the residual is not 100% consistent with the smoothness properties of the corresponding equation. For example, for the Poisson equation, the problem is selected in such a way the solution is analytic. However, for example, if the zero boundary conditions are enforced, and right hand side is all ones, the solution will have singularities. Thus, the main challenge would be the case when solution does have the singularities (and it will have it in many practical cases). The L2-norm then is not the right functional for the solution to exist, not to say about the higher-order derivatives. So, these functionals are not motivated by the theory of the solution of PDEs, but are rather focused on much smoother solution.
    - Convergence. There are quite a few papers on the convergence of DNN approximations to solution of PDEs. The presented methods might have converged to a local minimum. An important reference is the paper by Yarotsky D. Error bounds for approximations with deep ReLU networks. Neural Networks. 2017 Oct 1;94:103-14.